# Optoribogenetic control of regulatory RNA molecules

Sebastian Pilsl[1], Charles Morgan[1], Moujab Choukeife[1], Andreas Möglich [2] & Günter Mayer [1,3✉]

Short regulatory RNA molecules underpin gene expression and govern cellular state and physiology. To establish an alternative layer of control over these processes, we generated chimeric regulatory RNAs that interact reversibly and light-dependently with the light-oxygen-voltage photoreceptor PAL. By harnessing this interaction, the function of micro RNAs (miRs) and short hairpin (sh) RNAs in mammalian cells can be regulated in a spatio-temporally precise manner. The underlying strategy is generic and can be adapted to near-arbitrary target sequences. Owing to full genetic encodability, it establishes optoribogenetic control of cell state and physiology. The method stands to facilitate the non-invasive, reversible and spatiotemporally resolved study of regulatory RNAs and protein function in cellular and organismal environments.

[1] Life and Medical Sciences (LIMES), University of Bonn, Gerhard-Domagk-Str.1, 53121 Bonn, Germany. [2] Lehrstuhl für Biochemie, Photobiochemie, University of Bayreuth, Universitätsstraße 30, 95440 Bayreuth, Germany. [3] Center of Aptamer Research & Development, University of Bonn, Gerhard-Domagk-Str. 1, 53121 Bonn, Germany. ✉email: gmayer@uni-bonn.de

Short regulatory RNA molecules such as endogenous micro RNAs (miR) or synthetic short hairpin RNAs (shRNA) are essential mediators of gene expression[1–3]. They interact with defined complementary sites in the untranslated (UTR) or the coding regions of mRNA molecules, upon which translation is either inhibited or the mRNA is hydrolyzed. Regulatory RNAs have become indispensable in the biosciences for the validation of gene or protein function in cells and in vivo[4]. Although the on-demand control of mRNA translation has been achieved at the levels of mRNA stability and ribosome processing, e.g., by introducing aptazymes or aptamers in the UTRs[5–9], the direct control of the function of short regulatory RNAs, ultimately in a spatiotemporal manner, remains challenging. At the same time, it is highly demanded, as it would offer programmable, modular and generalizable control of target gene expression on the post-transcriptional level[10–12]. To this end, small-molecule-responsive siRNAs whose function can be controlled by theophylline or tetracycline[13] or conditional expressions systems of shRNAs[14] have been reported. These approaches extend towards the transcriptional regulation of miRs[15] or to aptazymes that control miR maturation in response to small molecules[16]. Atanasov et al. constructed *pre*-miR variants that functionally depend on the presence of doxycycline, mediated by a TetR-responsive aptamer[17], which has been previously used in combination with the theophylline aptamer to control transcription[18]. Besides these strategies, modalities to sequester miRs[19–21] or to inhibit their function by small molecules[22] were developed and applied in cell culture and in vivo. Most of these approaches rely on the exogenous addition of small molecules, which per se might interfere with other biological processes, have limited availability and stability in vivo, suffer from diffusional spread, and are of restricted reversibility[23]. To overcome certain of these limitations, light-dependent control of regulatory RNA has also been described[24–26], but the pertinent approaches invariably require chemical synthesis and the exogenous addition of the modified RNAs to biological systems. By contrast, entirely genetic approaches to gain spatiotemporal control over regulatory RNA function remain elusive but are highly desirable, as they would offer a plethora of applications to precisely and reversibly control gene expression and downstream processes.

Here, we devise a fully genetically encodable, generic approach that achieves light-dependent control of *pre*-miR and shRNA activity. We construct chimeric RNA molecules consisting of mature miR and siRNA sequences conjoined with an RNA aptamer that binds to the light-oxygen-voltage (LOV) photo-receptor PAL in a light-dependent manner[27,28]. The chimeric RNAs enable the spatiotemporal control of short regulatory RNA function in mammalian cells, as we showcase for the light-dependent control of gene expression and cell-cycle progression. This hitherto unavailable modality establishes a versatile RNA control system for analyzing various protein and miR functionalities in a reversible, spatiotemporally resolved, and non-invasive manner, and with full genetic encoding. Owing to the modularity of the chimeric RNAs, the technology readily applies to near-arbitrary shRNAs.

## Results

**PAL-mediated regulation of *pre*-miR activity.** Our design of light-responsive *pre*-miRs anticipates an altered processivity of short regulatory RNAs by Dicer owing to light-activated binding of the LOV receptor PAL to the apical loop domain[28–30]. To implement this design, we embedded the cognate aptamer domain of PAL in the apical loop of short regulatory RNAs. We hypothesized that thereby regulatory RNA function can be controlled by blue light (Fig. 1).

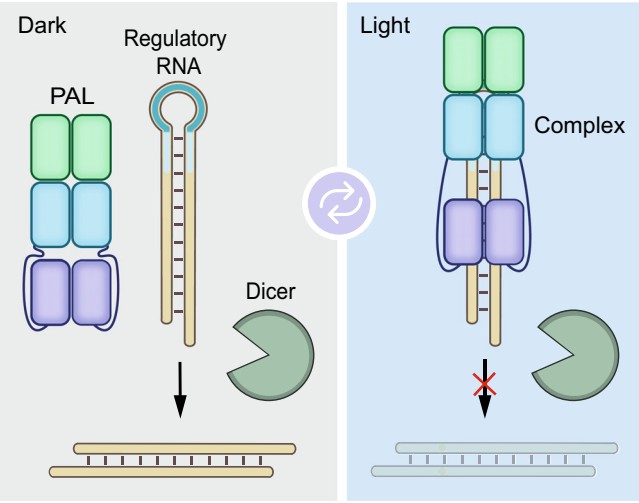

**Fig. 1 General design of light-dependent regulatory RNAs.** The PAL protein reversibly binds to its cognate RNA aptamer (highlighted in blue) embedded in the apical loop domain of a regulatory RNA (highlighted in light orange) in the light and thereby influences regulatory RNA function.

We generated *pre*-miR variants by replacing the apical loop domain with the PAL-binding RNA aptamer 53 (Fig. 2a, Supplementary Table 1). Notably, the RNA aptamer 53 interacts preferentially with the light-adapted state of PAL and to a much lesser extent with its dark-adapted conformation (Supplementary Fig. 3a, b). We first generated aptamer-modified variants of *pre*-miR-21 (SHA, Fig. 2a) and analyzed them in reporter gene assays that employ the expression of secreted *Metrida* luciferase or of enhanced green fluorescent protein (eGFP) with miR-21 target sites embedded in the 3′-UTRs of the respective mRNA (Supplementary Fig. 1a, b, 2)[31]. As controls, we constructed *pre*-miR-21 variants that bear a single point mutant (G11C) within the PAL aptamer that renders them binding-incompetent (SHC, SHD), a non-functional miR-21 domain (SHB, SHD)[32], or with both domains altered (SHD, Fig. 2a). Interaction experiments in vitro revealed light-dependent binding of SHA and SHB to PAL, similar to the parental aptamer (53), whereas *pre*-miR variants with mutated aptamer domains (SHC, SHD) did not bind (Supplementary Fig. 3a, b). For all experiments, a transgenic HEK293 cell line stably expressing mCherry-PAL (HEK293PAL) at an average concentration of 1 μM was used (Supplementary Fig. 4). The *pre*-miR-21 variants were transcribed under the control of the U6 promoter from plasmids[33] co-transfected with the luciferase reporter. Whereas SHA supressed luciferase expression in darkness, irradiation with blue light (λ = 465 nm) induced reporter gene expression by 4.4-fold to 27% of the maximal value (Fig. 2b, d). Replacing either the miR-21 domain with a non-targeting RNA (SHB) or the aptamer domain by a non-binding point mutant (SHC) resulted in a loss of light-regulation (Fig. 2b, d, Supplementary Fig. 5). Likewise, the *pre*-miR21 variant having both RNA domains altered (SHD) neither suppressed gene expression nor showed any light dependency (Fig. 2b, c). Analogous results were obtained for the eGFP reporter gene (Fig. 2d, e, Supplementary Figs. 7, 8, for details on eGFP gating strategy see Supplementary Fig. 6a), in that SHA inhibited expression in darkness, whereas a 4.4-fold induction of eGFP was observed in light (Fig. 2d, e). SHB and SHD did not inhibit eGFP expression, whereas SHC did, and none of the three variants exhibited light dependency (Fig. 2d). Intrinsic levels of argonaute 2 (AGO2) have been shown to limit RNA silencing efficiency[34]. Therefore, we co-expressed AGO2 and observed a more pronounced inhibition of eGFP expression by SHA and

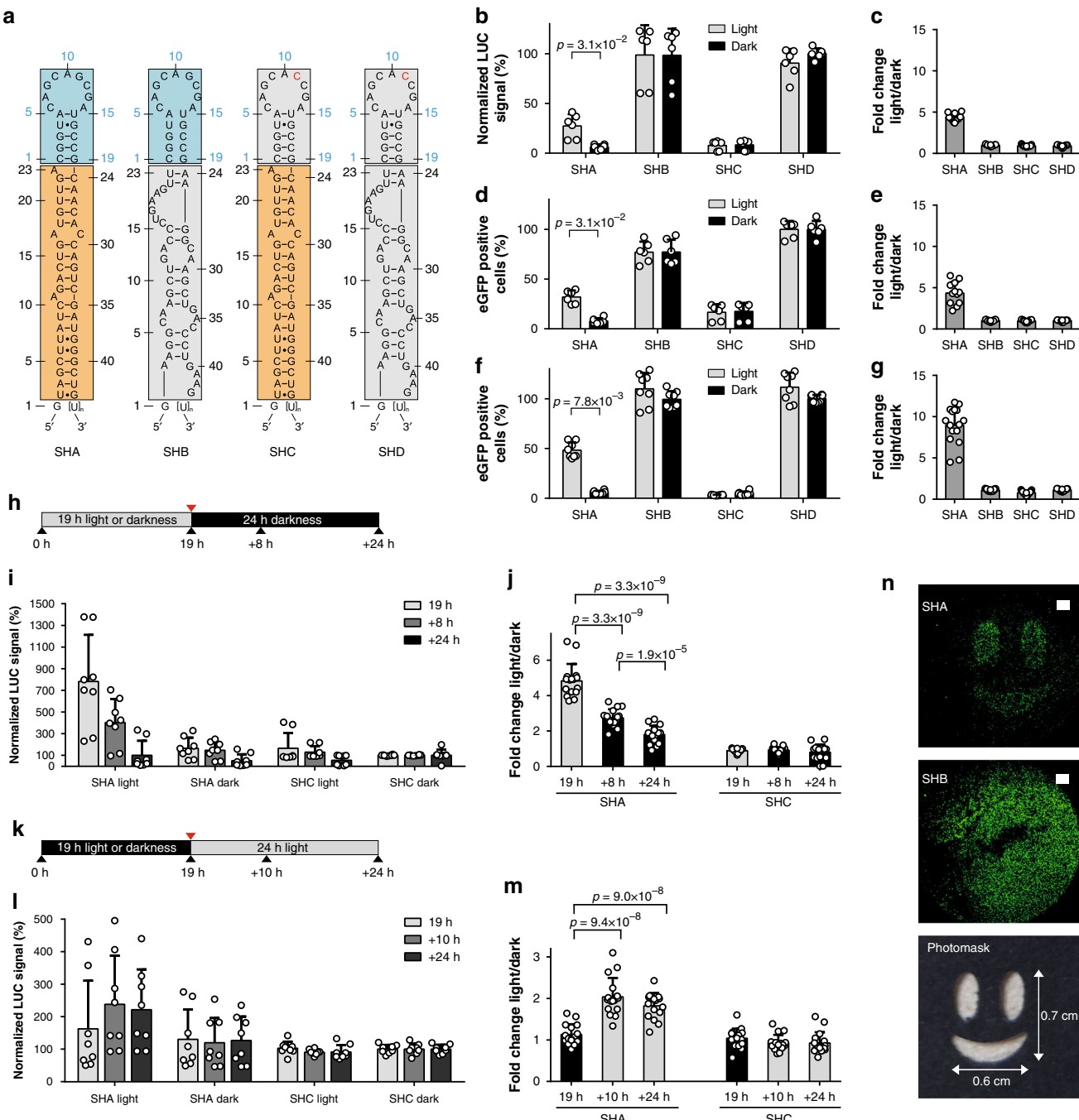

**Fig. 2 A *pre*-miR21-aptamer chimera enables light-control of gene expression. a** Schematic representation of the *pre*-miR21 variants and corresponding controls. Blue boxes: aptamer domain, orange boxes: miR21 domain, gray boxes: aptamer point mutant or control miR. **b** Luciferase expression after transfection of the indicated *pre*-miR21 variants. Values are normalized to SHD incubated in darkness. **c** Fold changes calculated from light *vs.* dark conditions from (**b**). **d** Number of cells expressing eGFP after transfection of the indicated *pre*-miR21 variants. Values are normalized to SHD incubated in darkness. **e**, Fold changes calculated from light *vs.* dark conditions from (**d**). **f** Number of cells expressing eGFP in the presence of elevated levels of AGO2 and after transfection of the indicated *pre*-miR21 variants. Values are normalized to SHD incubated in darkness. **g** Fold changes calculated from light *vs.* dark conditions from (**f**). $N = 6$ (**b**, **d**), 8 (**f**), 12 (**c**, **e**) or 16 (**g**). **b**–**g** Each biologically independent experiment was performed in duplicates. **b**, **d**, **f** Gray bars: light conditions, black bars: dark conditions. **c**, **e**, **g** Dark gray bars: fold changes. **b**–**f** Wilcoxon two-sided signed-rank test was used for statistical analysis as a paired observation was assumed. **h** Illumination protocol applied in (**i**) and (**j**). **i** Luciferase expression level of cells expressing SHA or SHC. Shown are normalized values to SHC in darkness. **j** Fold changes calculated from light *vs.* dark conditions from (**i**). **k** Illumination protocol applied in (**l**) and (**m**). **l** Expression level of luciferase of cells expressing SHA or SHC. Shown are normalized values to SHC in darkness. **m** Fold changes calculated from light vs. dark conditions from (**l**). **b**–**m** Values are means ± s. d. $N = 8$ (**i**, **l**) or 16 (**j**, **m**). Each biologically independent culture was performed in duplicates. **j**–**m** Two-sided Mann–Whitney *U* test was used for statistical analysis as an unpaired observation was assumed. **n** Spatial patterning of eGFP expression after transfection with SHA (top panel) or SHB (middle panel). Irradiation was done on cells covered with a photomask (bottom panel); white bars: 1000 μm. Source data for (**b**–**g**, **i**, **j**, **l**, **m**) are provided as a source data file.

SHC in darkness (Fig. 2f, g, Supplementary Figs. 9, 10). Irradiation induced eGFP expression in the cells harboring SHA by 9-fold (Fig. 2g). By contrast, experiments using SHB, SHC, and SHD did not reveal any light dependency (Fig. 2f, g).

We next assessed the reversibility of the approach using the luciferase reporter system. To this end, HEK293PAL cells harboring SHA were incubated for 19 h under blue light (Fig. 2h-j, Supplementary Fig. 11). Subsequently, the cells were kept in darkness for a further 24 h. An increase of luciferase activity in the cell culture supernatants was observed after 19 h in light and a reduction when cells were kept in the dark afterwards (Fig. 2i, j). In turn, cells kept first in darkness did not reveal luciferase expression (Fig. 2k-m, Supplementary Fig. 12), but luciferase activity was detected when cells were subsequently exposed to light conditions (Fig. 2l, m). Cells having SHC did not reveal light-dependent luciferase expression (Fig. 2i, j, l, m, Supplementary Figs. 11, 12). We also demonstrated spatial control of reporter gene expression using a photomask on HEK293PAL cells during irradiation (Fig. 2n). Expression of SHA resulted in eGFP expression predominantly in light-exposed areas, whereas eGFP expression was observed independently of the irradiation status in the presence of SHB (Fig. 2n).

To better characterize the processed miRs, we analyzed them by 3′ miR-RACE (rapid amplification of cDNA ends). Compared to reported natural pre-miR-21, we observed altered processing of SHA at the 3′end of miR-21-5p (Supplementary Table 2). We attribute this observation to using the U6 promotor for pre-miR-21 expression which requires an additional G-nucleotide for efficient transcription and, thus, induces altered Dicer processing[35].

**PAL-mediated regulation of shRNA activity.** We next investigated whether the PAL-aptamer system can also be applied to shRNA molecules in a more generic manner to thereby enable versatile optogenetic control of RNA interference[36]. Initially, we constructed two shRNAs (SH1, SH2) that target different sites within the eGFP mRNA coding region (Supplementary Fig. 1c) and conjoined them with the PAL aptamer (Fig. 3a, b). The expression of eGFP in HEK293PAL cells harboring SH1 or SH2 was light-responsive, with SH1 being more efficient in eGFP suppression in the dark (Fig. 3c, d, Supplementary Figs. 13-16). As a control we used the miR-21-targeting SHA (Fig. 2a), which did not inhibit eGFP expression (Fig. 3c, d, Supplementary Figs. 13–17, for details on eGFP gating strategy see Supplementary Fig. 6b), as the miR target site is absent in the reporter mRNA employed in this experiment (Supplementary Fig. 1c). Structural variations of one or two nucleotides surrounding the Dicer cleavage site are common motifs found in natural pre-miRs and shRNAs[37]. These motifs alter the accuracy of shRNA processing and, thus, gene silencing efficacy[38]. We hence extended our study towards examining the impact of the nucleotides' identity in the hinge region that connects the siRNA with the aptamer domain on shRNA performance. To this end, we designed 8 variants with single nucleotide bulges in the hinge region of SH1 (Fig. 3d) located either up- (SH3, SH4, SH5, SH6) or downstream (SH7, SH8, SH9, SH10) of the aptamer domain (Fig. 3e). All variants demonstrated light-dependent induction of eGFP expression but with varying efficiency (Fig. 3f). An upstream C (SH6) or a downstream G (SH9) nucleotide, relative to the aptamer domain, revealed the lowest expression, upstream A (SH3) or G (SH5) nucleotides or downstream A (SH7), U (SH8) or C (SH10) nucleotides exhibited very similar properties. Likewise, the suppression efficiency of shRNAs in the dark varied among the constructs (Supplementary Figs. 13-15). The observed fold changes of eGFP expression (light vs. dark) are comparable

across all shRNAs with SH9 having the lowest induction rate (Fig. 3g). In turn, an upstream U nucleotide (SH4) revealed similar fold changes (Fig. 3g) but a higher level of light-induced eGFP expression (Fig. 3f). Therefore, we chose A and U residues as representatives in the SH2 hinge region variants and included G (SH15) as a less efficient control. Single nucleotides inserted into the hinge regions of SH2 led to an improved eGFP knockdown in the dark (Fig. 3h, i, Supplementary Fig. 15), and all variants remained light-responsive. An adenine (SH11) or uridine (SH12) nucleotide at the hinge region upstream of the aptamer domain led to the highest number of eGFP-positive cells (Fig. 3i), and SH14 revealed the strongest increase of eGFP expression upon irradiation (15.3. fold) (Fig. 3j). SH16 with a mutated aptamer domain (G11C) did not reveal light-induced eGFP expression (Fig. 3h, i). Likewise, light-dependent induction of eGFP expression was also evident from fluorescence microscopy studies for the shRNA variants (Fig. 3k, Supplementary Fig. 17) SH1 (original hinge region), SH3 (intermediate performance), SH4 (highest number of eGFP positive cells when incubated in light), SH12 (second highest number of eGFP positive cells when incubated in light), and SH14 (highest light vs. dark fold change). In vitro binding studies verified light-dependent interaction with PAL of shRNA variants with engineered hinge regions (Supplementary Fig. 3a, b), indicating that these variations do not directly interfere with PAL binding but affect shRNA processivity. Of key importance, these findings testify to the modular design of the underlying chimeric RNAs and indicate that the domains for PAL-binding and mRNA-targeting are non-overlapping. As a corollary, we reasoned that near-arbitrary targeting domains should be accommodable with our technology.

**Optoribogenetic control of cell-cycle progression.** We hence extended our approach to regulating the expression of endogenous proteins via shRNAs. We chose cyclin B1 and CDK1 as targets, as they are both essential for the transition from the gap-2 ($G_2$) to the mitosis (M) phase of the cell cycle[39]. Variations of the expression levels of cyclin B1 and CDK1 have phenotypic consequences and alter the distribution of cells in different stages of the cell cycle[40,41]. First, we generated shRNAs targeting cyclin B1 with varied hinge nucleotides, having either an adenine (SHCB1) or uridine (SHCB2) upstream or uridine (SHCB3) downstream of the aptamer domain (Fig. 4a). HEK293PAL cells having the shRNAs SHCB1-3 in darkness (i.e. cyclin B1 knockdown condition) accumulated in the $G_2$/M phase (Fig. 4b, Supplementary Figs. 18, 19). Upon irradiation, the number of cells in $G_2$/M phase was significantly reduced in cells having SHCB1 (Fig. 4b, Supplementary Figs. 18, 19), indicating the recovery of normal cell-cycle propagation. By contrast, propagation was not recovered upon irradiation for the binding-incompetent aptamer variants of SHCB1 (G11C, SHCB1m) (Fig. 4b). SHCB2 and SHCB3 did not affect cell-cycle propagation when irradiated (Fig. 4b). Based on these results, we constructed PAL-dependent shRNA variants of CDK1 having an adenine nucleotide in the hinge region upstream of the aptamer (SHCDK1, Fig. 4c). HEK293PAL cells having the shRNAs SHCDK1 in darkness also accumulated in the $G_2$/M phase (Fig. 4d, Supplementary Figs. 18, 19). Upon irradiation, the number of cells in $G_2$/M phase was significantly reduced (Fig. 4d, Supplementary Figs. 18, 19). Cells having the PAL-binding deficient mutant shRNA SHCDK1m accumulated in the $G_2$/M phase irrespective of the irradiation status (Fig. 4b, d, Supplementary Figs. 18, 19). No accumulation of cells in the $G_2$/M phase was observed when cells expressed the non-targeting SH3 or were untreated (Supplementary Fig. 20).

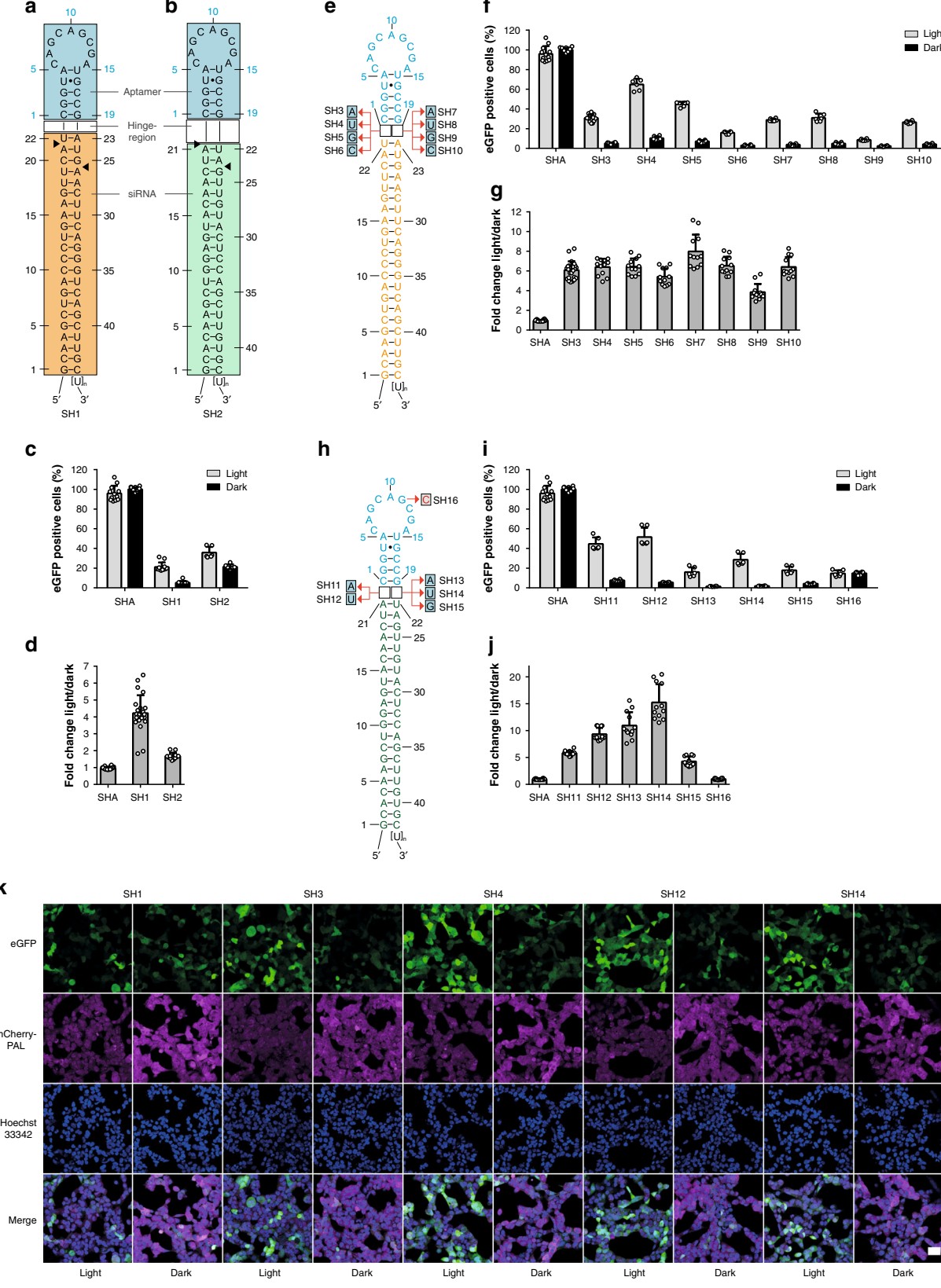

However, a slight accumulation of cells in $G_2/M$ phase was observed upon irradiation (Supplementary Fig. 20), most likely because of secondary irradiation effects on cells[42]. SHCB1 or SHCDK1 led to a decrease of cyclin B1 and CDK1 expression, respectively, which was reversed by irradiation (Fig. 4e–g, Supplementary Fig. 21). Variants of the shRNAs deficient for

PAL-binding (SHCB1m, SHCDK1m) suppressed protein expression independently of light (Fig. 4e–g). The non-targeting shRNA SH3 (Fig. 3b) did not affect cyclin B1 and CDK1 expression, the expression levels of both proteins were similar those when cells were untreated (Fig. 4e–g, Supplementary Fig. 21). Of note, the shRNA variants targeting cyclin

**Fig. 3 Design of shRNAs for the light-dependent expression of eGFP.** Two different siRNA sequences SH1 (orange box, **a**) and SH2 (green box, **b**) targeting eGFP mRNA were conjoined with the PAL aptamer (blue boxes) as apical loop domains. Black arrows indicate a putative preferential dicer cleavage site[44]. **c** Number of cells expressing eGFP after transfection of SH1 or SH2. Values are normalized to SHA (Fig. 2a) in darkness. **d** Fold changes calculated from light vs. dark conditions from (**c**). **e** Single nucleotide permutations of the hinge region in SH1 and their impact on eGFP expression and light dependency (**f**). Values are normalized to SHA in darkness. **g** Fold changes calculated from light *vs.* dark conditions from (**f**). **h** Single nucleotide permutations of the hinge region in SH2 and their impact on eGFP expression and light dependency (**i**). Values are normalized to SHA in darkness. **j** Fold changes calculated from light *vs.* dark conditions from (**i**). **k** Fluorescence microscopy images of cells transfected with the indicated shRNA variants. Cells were incubated under either light or dark conditions. Scale bar: 40 μm. **c**, **f**, **i** $N = 12$ (SHA, SH1, SH3) or 6 (all others). **d**, **g**, **j** $N = 24$ (SHA, SH1, SH3) or 12 (all others). **c**–**k** Each biologically independent experiment was performed in duplicates. Gray bars: light conditions, black bars: dark conditions. Dark gray bars: fold changes. Values are means ± s.d. Source data for (**c**, **d**, **f**, **g**, **i**, **j**) are provided as a source data file.

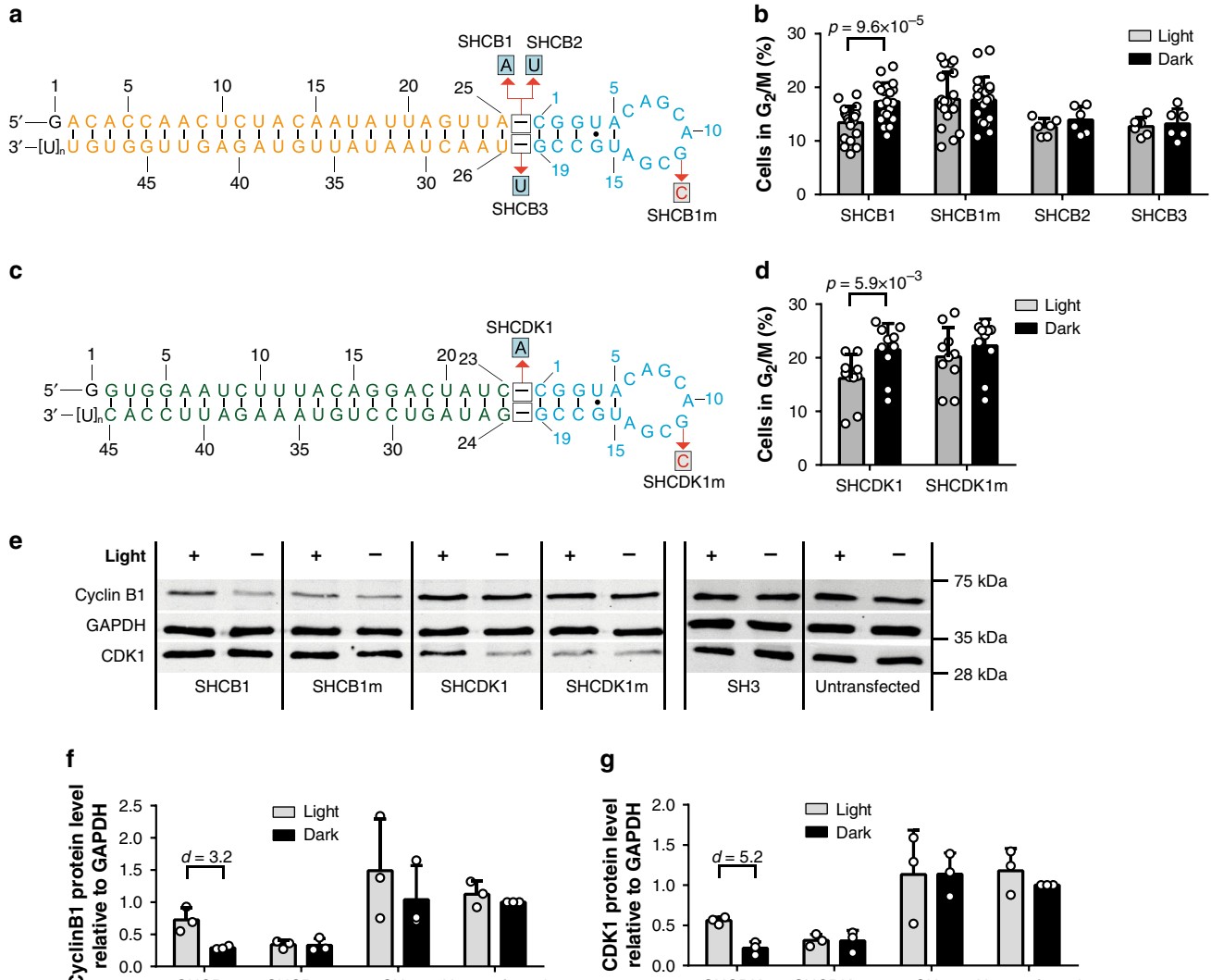

**Fig. 4 Optoribogenetic control of the mammalian cell cycle. a** shRNA variants used to control cyclin B1 gene expression. Blue: aptamer domain; orange: siRNA domain. **b** Percentages of HEK293PAL cells in $G_2/M$ phase of the cell cycle when transfected with indicated shRNAs targeting cyclin B1. **c** shRNA variants used to control CDK1 gene expression. Blue: aptamer domain; green: siRNA domain. **d** Percentages of HEK293PAL cells in $G_2/M$ phase of the cell cycle when transfected with indicated shRNAs targeting CDK1. **b** $N = 20$ (SHCB1, SHCB1m) or 6 (SHCB2, SHCB3). **d** $N = 10$. **b**, **d** Each biologically independent experiment was performed in duplicates. **b** The identity of SHCB1 and SHCB1m was blinded and double-blinded in one experiment, each. **d** The identity of SHCDK1 and SHCDK1m was blinded and double-blinded in one experiment, each. **b**, **d** Wilcoxon two-sided signed-rank test was used for statistical analysis. **e**, Representative western blot image showing cyclin B1, CDK1 and GAPDH protein expression after transfection with the indicated shRNAs (for complete blots see Supplementary Fig. 21). **e**, $N = 4$. **e** One biologically independent experiment was performed in duplicates, all others once. **f**, **g** Quantification of cyclin B1 and CDK1 protein levels using pixel densitometry. **f**, **g** $N = 3$. **f**, **g** All biologically independent experiments were performed once. **f**, **g** Cohen's $d$ effect size was used for statistical analysis. Values were normalized to non-transfected cells incubated in darkness (Untransfected). **b**, **d**, **f**, **g** Gray bars: cells incubated under light conditions, black bars: cells incubated under dark conditions. Values are means ± s.d. Source data for (**b**, **d**, **f**, **g**) are provided as source data file.

B1 did not affect CDK1 expression and vice versa (Fig. 4e, Supplementary Fig. 21).

## Discussion

In conclusion, we demonstrate the fully genetically encodable light-control of miR and shRNA molecules in mammalian cells. The approach utilizes an aptamer that under blue light binds tightly and specifically to the photoreceptor protein PAL, and this interaction was shown to impact miR and shRNA function in regulating gene expression. We thus created an encoded on-switch, complementing a previously reported off-switch in which the PAL aptamer was embedded directly in the 5′UTR of mRNAs[28]. By offering full genetic encodability, reversibility, and noninvasiveness combined with a small genetic footprint (ca. 1.1 kb), our approach transcends previous approaches for controlling regulatory RNA activity. Specifically, these features distinguish our method from ligand-gated techniques that invariably rely on the exogenous addition of specific compounds, thus abolishing full genetic encoding and limiting their application scope. Our method rivals CRISPR/Cas9-based approaches in its ready adaptability to target sequences through variation of the modular chimeric RNA. The technology thus unlocks optogenetic control of near-arbitrary gene products at the post-transcriptional level and expands the optogenetic toolbox. It should be noted that engineered *pre*-miR molecules might be more difficult to be approached, as genetic manipulation is required. Notably, the shRNA-based approach operates dominantly and can hence be used in wild-type cellular backgrounds, thus obviating the laborious construction of transgenic lines. To facilitate adoption of the technology, we investigated in detail sequence determinants affecting the efficiency of light regulation. We demonstrate that single nucleotide variations in the hinge region connecting the miR/siRNA and the aptamer domains impact on regulatory RNA function and allow its fine-tuning, with an up to 15-fold change in protein expression presently. Although we observed a preference for A and U nucleotides of the best-performing shRNAs, we recommend testing all canonical nucleotides (G,U,A, and C) at the hinge region, upstream and downstream of the aptamer domain to identify the most suitable variant. The induction of gene expression in light still reveals residual shRNA activity, depending on the regulatory RNA and target mRNA used. Further optimization might increase the dynamic range of the system, e.g., by varying the identity of the target region in a mRNA and, thus, the shRNA sequence. Besides light dependency, we also demonstrate spatial and temporal regulation and the suitability of the system to control endogenous proteins and cellular behavior, exemplified by controlling the cyclin B1 and CDK1 protein expression. This optoribogenetic approach extends to various shRNA and miR molecules for the investigation of dynamic biological processes by light, e.g., the relationship of proliferation and differentiation of neuronal stem cells, which depends on the progression of the cell cycle[43]. Additionally, optoribogenetic approaches may contribute to the understanding of dynamic micro RNA and protein functions that remain challenging to be resolved with the currently available methodologies.

## Methods

**Molecular biology.** All oligonucleotides were purchased from Ella Biotech, Planegg, Germany. Plasmid pIRESneo-FLAG/HA Ago2 was kindly provided by Thomas Tuschl. The plasmids pmCherry-C1, pMetLuc2-Control and peGFP-N1 were purchased from Takara Clontech. All miR and shRNAs described herein were cloned in the pSilencer 2.0-U6 plasmid backbone using restriction cloning. Corresponding miR and shRNA sequences are listed in Supplementary Table 1. The identity of all constructs was confirmed by Sanger DNA sequencing (Eurofins).

**PAL protein purification and RNA:PAL interaction assay.** For protein expression, the plasmid pET-28c-PAL was transformed into ArcticExpress *e. coli* cells

(DE3, Agilent). Bacteria were grown in lysogeny broth (LB) medium supplemented with 50 µg mL$^{-1}$ kanamycin, 20 µg mL$^{-1}$ gentamycin and 50 µM riboflavin at 37 °C at 120 r.p.m. until an optical density (OD) at 600 nm of 0.6 was reached, at which point expression was induced by addition of 1 mM isopropyl β-D-1-thiogalactopyranoside (IPTG). Incubation continued at 16 °C and 120 r.p.m. for 66 h, after which cells were harvested and lysed by ultrasound. The lysate was cleared by centrifugation and applied to an immobilized nickel ion affinity column (Marcherey Nagel). Protein was eluted in buffer A (50 mM Tris/HCl, 200 mM NaCl, 200 mM imidazole, 10% w/v glycerol, pH 7.6). Protein was dialyzed into buffer B (12 mM HEPES/KOH, pH 7.2, 135 mM KCl, 10 mM NaCl, 10% w/v glycerol). Identity of the protein was verified by polyacrylamide gel electrophoresis (PAGE). Protein concentration was determined by absorption spectroscopy using an extinction coefficient of 12,500 M$^{-1}$ cm$^{-1}$ at 447 nm.

In vitro transcription was performed using T7 RNA polymerase. DsDNA template coding for shRNA sequences was modified upstream by implementing T7 promoter sequence and two additional guanine residues (bold) after the transcription start site to ensure transcription (TAATACGACTCACTATA**GG**, underlined G: transcription start site). DNaseI (Roche) digested RNA was purified by PAGE and recovered by electroelution (8 M ammonium acetate, 45 min, 150 V). PAGE and electroelution were performed using TBE buffer (100 mM Tris, 100 mM boric acid, 2 mM ethylenediaminetetraacetic acid, pH 8.0).

For RNA:PAL interaction assays, PAL protein was biotinylated with a four-fold excess of EZ-Link Sulfo-NHS-LC-Biotin according to the manufacturer's instructions (Thermo Fisher Scientific) and coupled to streptavidin-coated wells of a plate (Pierce Streptavidin Coated Plates, Black, 96-Well, Thermo Fisher Scientific). Wells were washed three times with 200 µl buffer B. Subsequently, 100 µl of 1.5 µM biotinylated PAL in buffer B was added and the coupling was performed in darkness over night at 4 °C. Afterwards, wells were washed three times with 200 µl buffer B. In vitro transcribed *pre*-miR or shRNA constructs were incubated with immobilized PAL at the indicated concentrations in 100 µl buffer B for 30 min at 25 °C under light ($\lambda_{max}$ = 465 nm, 2.15 W cm$^{-2}$) or in darkness, followed by three washing steps with 200 µl buffer b for 3 min each. Fluorescence detection was performed by adding 150 µl RiboGreen reagent (Quant-iT RiboGreen RNA Reagent, Thermo Fisher Scientific), diluted 500-fold in 1× TE buffer (10 mM Tris/HCl, pH 7.5, 1 mM ethylenediaminetetraacetic acid) to each well. After 1 h incubation in darkness, fluorescence intensity was measured on a Tecan Ultra plate reader (Tecan) at excitation and emission wavelengths of 500 and 525 nm, respectively. Results were normalized to full length aptamer (53) incubated under light conditions.

**LED array.** Blue light was administered to cells in pulses (30 s light on, 30 s light off) by a custom LED array (illuminance at 106 µW cm$^{-2}$) with $\lambda_{max}$ = 465 nm. The LED array was powered using a custom-built microcontroller. Cells were exposed to light immediately after transfection until they were subjected to further investigation.

**Light plate apparatus (LPA).** A custom-made replicate of the LPA was produced by Hanns–Martin–Schmidt and used for the photomask experiment.

**Working with mammalian cell lines.** HEK293 cells (CLS Cell Lines Service) were cultured in DMEM medium (high glucose, GlutaMAX), supplemented with 1% non-essential amino acids (NEAA), 1% Sodium Pyruvate (Thermo Fisher Scientific) and 10% fetal calf serum (FCS, Sigma-Aldrich), at 37 °C in a humidified 5% CO$_2$ atmosphere and were passaged every 2–3 days. HEK293PAL cells were cultured 1 week before and after cell sorting with 1% penicillin/streptomycin and in presence of Geneticin (G418, 400 µg mL$^{-1}$, Thermo Fisher Scientific). Mycoplasma testing was performed every three months using PCR detection (Minerva biolabs).

**Transient transfection.** Cells were transfected 24 h after incubation in darkness in 500 µl DMEM supplemented with 1% NEAA and 1% Sodium Pyruvate. 1.5 µL Lipofectamine2000 (Thermo Fisher Scientific) was used per well in a 24-well plate format. For transfection experiments involving AGO2, cells were transfected with 250 ng plasmid DNA (e.g., pSilencer, AGO2 and pEGFP-N1 plasmid at a mass ratio of 2:2:1). For reporter assays without AGO2 overexpression, cells were transfected with 500 ng plasmid DNA (e.g. pSilencer and pEGFP-N1 or PMetLuc2-Control plasmid at a mass ratio of 100:1). For transfections of shRNAs targeting intrinsic mRNAs, 500 ng of the corresponding pSilencer plasmid variant was transfected. Plasmids and Lipofectamine2000 were each diluted in 50 µl Opti-MEM (Thermo Fisher Scientific) per well and incubated 5 min before mixing at room temperature. After 20 min further incubation at room temperature, 100 µl transfection mix was added per well. Four hours after transfection, 60 µl FCS were added per well.

**Flow cytometry analysis.** Flow cytometry was performed on a BD FACSCanto II (BD Biosciences). Data was processed using FlowJo version 9.6.3 software. Cells were initially gated with SSC and FSC channels for cell debris and single cell populations. For measuring eGFP, cells were additionally gated using the fluorescein isothiocyanate (FITC) channel to determine eGFP positive cells. For cell-cycle measurements, cells were additionally gated using Phycoerythrin (PE) channel to determine the

percentage of cells in the different cell-cycle phases. At least 30.000 cells were analyzed from each sample. eGFP and propidium iodide were excited with a 488 nm laser and detected with a 530/30 or 585/42 filter set, respectively. mCherry was excited with a 633 nm laser and detected with a 660/20 filter set.

**Isolation of monoclonal cells.** For the generation of stable monoclonal HEK293 cell lines expressing mCherry-PAL (HEK293PAL), $10^6$ cells were seeded into each well of a 6 well plate and transfected by Lipofectamine2000 transfection on the following day using 2.5 µg plasmid DNA and 8 µl Lipofectamine2000. Four hours after transfection, the supernatant was discarded and cells were washed with PBS before the addition of 3 mL cell medium. After three days, the medium was supplied with 400 µg mL$^{-1}$ G418. One week before and after cell sorting cell medium was supplemented with 1% penicillin/streptomycin (Thermo Fisher Scientific). After five weeks of selection, single cells, which strongly express mCherry, were sorted via fluorescence activated cell sorting into a well of a 96-well plate. Cells were cultivated until 80% confluency was reached in a T-175 culture flask. Then, cells were frozen or used in further experiments. We did not notice any change in growth rate and cell fitness of the HEK293PAL cell line compared to HEK293 cells.

**Luciferase reporter assays in mammalian cells.** $1 \times 10^5$ HEK293PAL cells were seeded in two separate 24-well plates per well. After 24 h, cells were transfected according to the protocol described above. Transfected cells were incubated for 19 h in the presence of blue light using the LED array or in darkness.

For reversibility assays, one control plate was kept constantly in darkness. Another plate was incubated under varying light conditions after transfection as indicated by the time lines shown in Fig. 2h, k. Immediately before the light irradiation status was altered for one plate, cell medium was exchanged for both plates involved in the assay (plate incubated constantly in darkness and plate exposed to varying light conditions).

For the luciferase assay, 50 µl of the cell culture supernatant was transferred to wells of a white 96-well plate (LUMITRAC 200, Greiner). Five microliters of luciferase substrate dissolved in buffer according to the manufacturer's instructions (Ready-To-Glow Secreted Luciferase, Takara Clontech) was added and the reaction was incubated for 3 min at room temperature. The luminescence signal was measured using an EnSpire plate reader (PerkinElmer) with an integration time of 5 s.

For the static luciferase assay (Fig. 2b), values were normalized to control pre-miR transfection (SHD) incubated in darkness, where no influence on luciferase expression was expected. For the reversibility luciferase assay (Fig. 2i, l), values were normalized to aptamer point mutant pre-miR21 variant (SHC) incubated constantly in darkness, where no light dependency was expected. Normalization was performed to each time point.

**Photomask experiment.** $7.5 \times 10^4$ HEK293PAL cells were seeded in black 24-well plates with clear bottom (VisionPlate, 4titude). After 24 h, cells were transfected using the protocol for AGO2 overexpression as described in the transient transfection section. Then, the plate was mounted onto the custom-made LPA device and irradiated with 10 µW cm$^{-2}$ of constant light ($\lambda_{max} = 465$ nm). After 48 h, cells were analyzed by confocal laser scanning microscopy (LSM 710) using a 10×/0.45 objective and image concatenation (10% overlay). Imaging was performed at 37 °C. eGFP fluorescence was visualized as green color and image histograms were adjusted to 5/10 before.tiff picture export (Zen Black software, Zeiss). Image brightness was adjusted to + 150 and image sizes were adjusted to 300 × 300 pixels using Adobe Photoshop CS5 software.

**eGFP reporter assays in mammalian cells.** $1 \times 10^5$ HEK293PAL cells were seeded in two separate 24-well plates per well. After 24 h, transfection was performed as indicated in the transient transfection section. For the miR21-based reporter assays, the 3′UTR of peGFP-N1 bearing miR21 binding sites (Supplementary Fig. 1b) was used. Cells were incubated for 44 h in the presence of blue light using the LED array or in darkness. Then, the cell supernatant was aspirated, cells were washed and resuspended in PBS (25 °C). The subsequent analysis was performed using flow cytometry. The percentage of eGFP positive cells was normalized to control pre-miR transfection (SHD) incubated in darkness where no influence on eGFP expression was expected for miR experiments. For shRNA experiments, the percentage of eGFP positive cells was normalized to functional pre-miR21 transfection containing the PAL aptamer (SHA) and incubated in darkness as no influence on eGFP expression was expected due to the absence of miR21 binding sites in the 3′UTR sequence of the eGFP mRNA.

**Fluorescence microscopy of HEK293PAL cells.** $5 \times 10^4$ cells were seeded in black 24-well plates with clear bottom (µ-plate, ibidi). After 24 h, cells were transfected using the protocol for AGO2 overexpression as described in the transient transfection section. Cells were incubated for 44 h in the presence of blue light (465 nm, 106 µW cm$^{-2}$, 30 s pulses) or in darkness. Then, the supernatant was replaced by cell medium containing 5 µg mL$^{-1}$ Hoechst 33342. After 10 min incubation at 37 °C, the supernatant was replaced with cell medium only. Next, cells were analyzed by confocal laser scanning microscopy (LSM 710, Zen Black software, Zeiss) using a 20×/0.8 objective. We generated pictures comprising of 1528 × 1528 pixels with a resolution of 0.19 µm per pixel and a pixel dwell time of 2.11 µs for imaging of each

fluorophore. Fluorescence of mCherry (excitation (ex)/emission (em): 543/578–696 nm), Hoechst 33342 (ex/em: 405/410–494) and eGFP (ex/em: 488/494–574 nm) was monitored, respectively. Imaging was performed at 37 °C.

**Western blot analysis.** HEK293PAL cells were lysed 44 h after transfection in RIPA buffer (Thermo Fisher Scientific) containing 1 mM Phenylmethylsulfonyl fluoride. Lysates were cleared by centrifugation (14,000 $g$ for 15 min, 4 °C). Cleared lysates in Laemmli buffer were incubated at 95 °C for 5 min before loading on SDS-PAGE gels. Protein quantification was performed using the Pierce BCA protein assay kit according to the manufacturer's instruction (Thermo Fisher Scientific). 5 µg of protein per lane was loaded onto 10 or 12.5% SDS-PAGE gels and blotted in Transfer Buffer (2.5 mM Tris, 2% (w/v) glycine, 0.9 M urea) onto a nitrocellulose membrane (GE Healthcare Life Sciences) using a Bio-Rad Trans-Blot SD Semi-Dry Transfer Cell (BioRad) for 75 min at 20 V and 30 W. Membranes were blocked with TBS-T buffer (20 mM Tris/HCl, pH 7.6, 150 mM NaCl, 0.05% Tween 20 (v/v)) containing 5% BSA (AppliChem, Western Blot grade) under agitation at room temperature for 1 h. Blots were cut according to the protein ladder (Prestained Protein Ladder—Mid-range molecular weight (10–180 kDa), abcam) in a way that all target proteins can be individually incubated with the respective primary antibody (mouse anti-cdc2 (CDK1), Cell signaling POH1, #9116 (1:1000); mouse anti-GAPDH, Santa Cruz Biotechnology sc-47724 (1:4000); goat anti-Cyclin B1, R&D Systems AF6000 (1:1000)) at 4 °C overnight or at room temperature for 1 h in TBS-T containing 5% BSA. Detection was performed with IRDye 800CW goat anti-mouse (Li-cor 926–32210), donkey anti-goat (Li-cor 926–32214) and goat anti-rabbit (Li-cor 926–32211) at a dilution of 1:15000, respectively.

Antibody-stained blot pieces were arranged, and fluorescence images were acquired using an Odyssey Imaging System's (Li-cor) 800 nm channel (ex/em: 785/810 nm) to visualize bound 800CW secondary antibodies. Pixel densitometry of blot bands was performed using Fiji software (ImageJ) by creating rectangles of equal sizes for each sample lane followed by quantification of the area under the peak of each protein spot. Relative density was calculated by dividing values obtained from putative CDK1 or cyclin B1 protein bands through the values obtained from the putative GAPDH band of each lane. Relative density values were normalized to untransfected cells incubated in darkness.

**Cell-cycle assay.** $1 \times 10^5$ HEK293PAL cells per well were seeded in two separate 24-well plates. 24 h after seeding, transfection was performed according to the transfection protocol for shRNAs targeting intrinsic mRNAs as indicated in the transient transfection section. Cells were incubated for 44 h in the presence of blue light (465 nm, 106 µW cm$^{-2}$, 30 s pulses) or in darkness. 44 h after transfection, cells were fixed in 70% ice-cold Methanol in PBS and incubated for at least 30 min at 4 °C followed by RNAse A (50 µg mL$^{-1}$) and propidium iodide (PI, 50 µg mL$^{-1}$) treatment for 30 min at 37 °C under mild agitation. Subsequent analysis of cell-cycle distribution quantified by PI fluorescence per cell was performed using flow cytometry.

**mCherry quantification in HEK293PAL cells.** $1 \times 10^6$ HEK293PAL cells were lysed in 250 µl RIPA buffer (Thermo Fisher Scientific) containing 1 mM Phenylmethylsulfonyl fluoride (PMSF). Lysates were cleared by centrifugation (14,000 $g$ for 15 min, 4 °C). Generation of a mCherry standard curve and fluorescence measurements were performed using mCherry Quantification Kit (BioCat) according to the manufacturer's instruction with the exception that RIPA buffer containing 1 mM PMSF was used instead of mCherry Assay Buffer. For calculation of the concentration of mCherry-PAL a cellular volume of 4000 µm$^3$ was assumed and the cytoplasm covering 1/3 of this volume.

**Determination of 3′ ends of artificial miR21-5p.** $1 \times 10^5$ HEK293PAL cells were seeded in a 24-well plate. 24 h later, transfection with SHA and luciferase reporter plasmid was performed and cells were further incubated in darkness. 19 h after transfection total RNA extraction with TRIzol was performed according to the manufacturer's instructions (Thermo Fisher Scientific). 1.5 µg total DNase-treated RNA was poly-adenylated using Poly-A Polymerase according to the manufacturer's instructions (New England Biolabs). The reaction was purified by phenol/chloroform extraction followed by ethanol precipitation. Reverse transcription was performed using 1 µg poly-adenylated cDNA, Bioscript Reverse Transcriptase according to the manufacturer's instructions (Bioline, Supplementary Table 3). After another round of phenol/chloroform extraction followed by ethanol precipitation, PCR amplification was performed using Taq Polymerase and specific primers (Supplementary Table 3). Cloning was performed using TOPO-TA cloning kit according to the manufacturer's instructions (Thermo Fisher Scientific). Plasmids were isolated from individual clones using Plasmid DNA purification kit (Marcherey Nagel) and sent for sequencing (Eurofins).

**Calculation of fold changes.** Light-dependent fold changes were calculated by dividing values (crude values for luciferase assay, percentage of eGFP positive cells for eGFP assays) of samples incubated under light conditions through samples incubated in darkness (duplicates) to obtain four values for each independent experiment.

**Blinded experiments**. The identity of SHCB1, SHCB1m, SHCDK1, SHCDK1m and SH3 which are shown in Fig. 4b, d and Supplementary Figs. 18 and 20 were blinded and double-blinded in one experiment, each. For the blinded experiment, the identity of these PSilencer plasmids have been blinded before transfection by a second person. For the double-blinded experiment, another experimenter performed the assay that was blinded by a third person in a similar way. Identity of the samples was confirmed after data evaluation between the experimenter and the person who performed the blinding.

**Statistics and reproducibility**. Prism 6.01 (GraphPad Software, Inc.) was used to generate graphs and calculate $P$ values. For all statistical analysis, no gaussian distributions were assumed due to limited sample sizes. Wilcoxon two-sided signed-rank test was used to compare equally treated cell samples incubated under the indicated light conditions. Therefore, a paired observation was assumed. Two-sided Mann–Whitney $U$ test was used to compare light-dependent fold changes between differently treated groups (e.g. different time points). Therefore, an unpaired observation was assumed. Cohen's $d$ effect size was used to calculate effect sizes in Western Blot analysis derived from equally treated cell samples incubated under the indicated light conditions. In this case, sample size was too small to test for significance. Datasets are presented as mean ± s.d., if not otherwise stated.

**Reporting summary**. Further information on research design is available in the Nature Research Reporting Summary linked to this article.

## Data availability

Data are available in the main text, the Supplementary Information, or from the corresponding author upon request. Plasmids from Addgene (#10822) were used in this study. Source data are provided with this paper.

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

## Acknowledgements

This work was supported by funds from the European Union ERC ('OptoRibo', 615381) to G.M., and the German Research Council (grants MA3442/5-1 and 5-2 to G.M., and MO2192/6-1 to A.M.). We thank Prof. Gruss for critical commenting on the cell-cycle data. The FACS core facility of the University Hospital Bonn is

acknowledged for support in applying the gating strategy for the eGFP expression experiments.

## Author contributions

S.P. designed the shRNAs, developed and performed all PAL-dependent experiments in mammalian cells and wrote the manuscript. C.M. performed the interaction studies of RNA molecules with PAL, M.C. performed the blinded studies, A.M. conceived the project and discussed experiments, G.M. conceived the study, supervised, discussed, designed the experiments, and wrote the manuscript.

## Funding

## Competing interests

The authors declare no competing interests.
