## [Peer Review File · Nature Communications]

REVIEWERS' COMMENTS:

Reviewer #1 (Remarks to the Author):

This is a manuscript by Pilsel et al. reporting a new light-inducible and reversible platform to control the activity of small regulatory RNAs for enhanced spatiotemporal regulation. In this revised manuscript, authors have conducted additional experiments and revised the manuscript that addressed comments from the previous review and greatly strengthened the manuscript. I believe that this work will be suitable to be published in *Nature Communications* after some minor revisions:

1. As authors agreed that "studying native pre-miR function might be laborious and difficult to realize, although technically possible". It is obvious that this method is more suitable for applications in shRNA instead of miRNAs. This should be commented in the discussion section to help readers in deciding whether this is the right tool for their studies. The current abstract and manuscript read like the method applied equally well to both miRNAs and shRNAs.
2. Authors should also discuss in the manuscript regarding the high background of inhibition in the light state. Although authors can indeed induce further effects on top of this background, such background caused by introducing the system into cells already significantly perturb the endogenous environments (50-80% knockdown before changing the state). Maybe the reported method can achieve comparable knockdown levels as other methods, I don't think other methods have such high background effects. The question then depends on how the results will be interpreted – whether to compare the results between light and dark states or to compare with non-transfected cells.

Reviewer #1:

This is a manuscript by Pilsel et al. reporting a new light-inducible and reversible platform to control the activity of small regulatory RNAs for enhanced spatiotemporal regulation. In this revised manuscript, authors have conducted additional experiments and revised the manuscript that addressed comments from the previous review and greatly strengthened the manuscript. I believe that this work will be suitable to be published in Nature Communications after some minor revisions:

1. As authors agreed that “studying native pre-miR function might be laborious and difficult to realize, although technically possible”. It is obvious that this method is more suitable for applications in shRNA instead of miRNAs. This should be commented in the discussion section to help readers in deciding whether this is the right tool for their studies. The current abstract and manuscript read like the method applied equally well to both miRNAs and shRNAs.

Answer: We thank the reviewer for pointing this out. The following sentence was added to the main text (page 13, line 293): “It should be noted that engineered *pre*-miR molecules might be more difficult to be approached, as genetic manipulation is required.”

2. Authors should also discuss in the manuscript regarding the high background of inhibition in the light state. Although authors can indeed induce further effects on top of this background, such background caused by introducing the system into cells already significantly perturb the endogenous environments (50-80% knockdown before changing the state). Maybe the reported method can achieve comparable knockdown levels as other methods, I don't think other methods have such high background effects.

Answer: We thank the reviewer for this valuable comment and for clarity we added the following sentences to the revised version of the manuscript (page 13, line 303): “The induction of gene expression in light still reveals residual shRNA activity, depending on the regulatory RNA and target mRNA used. Further optimization might increase the dynamic range of the system, e.g., by varying the identity of the target region in a mRNA and, thus, the shRNA sequence.”

The question then depends on how the results will be interpreted – whether to compare the results between light and dark states or to compare with non-transfected cells.

Answer: We agree with the reviewer that data obtained from light-responsive constructs can be compared between light and dark states or compared to non-transfected cells. That is why we show light and dark states along with non-transfected cells in several experiments to reveal a comprehensive picture (see for example in **Fig. 4f,g** and **Supplementary Fig. 17,19,21**).